# Potential Use of Two Forest Species (*Salix alba* and *Casuarina glauca*) in the Rhizofiltration of Heavy-Metal-Contaminated Industrial Wastewater

Malek Bousbih [1,2], Mohammed S. Lamhamedi [3], Mejda Abassi [2], Damase P. Khasa [4] and Zoubeir Béjaoui [1,2,*]

1 Faculty of Sciences of Bizerte, University of Carthage, Jarzouna 7021, Tunisia
2 Laboratory of Forest Ecology (LR11INRGREF03), National Institute of Research in Rural Engineering, Water and Forests (INRGREF), University of Carthage, Hédi Elkarray Street, Elmenzah IV, BP 10, Ariana 2080, Tunisia
3 Direction de la Recherche Forestière, Ministère des Ressources Naturelles et des Forêts, 2700, Rue Einstein, Quebec, QC G1P 3W8, Canada
4 Centre for Forest Research and Institute for Systems and Integrative Biology, Université Laval, 1030 Avenue de la Médecine, Quebec, QC G1V 0A6, Canada
* Correspondence: zoubeir.bejaoui@fsb.ucar.tn

**Abstract:** The discharge of raw industrial wastewater (IWW) into ecosystems is a major environmental problem that adversely affects water quality, soil physicochemical properties, the food chain and, therefore, human health. Injection of treated IWW into irrigation and "fertigation" systems is an ecological, sustainable and economical approach for its appropriate disposal. Seedlings of two forest species (*Salix alba*, *Casuarina glauca*) were grown hydroponically and subjected to 25% diluted IWW and control (tap water) treatments for 35 days. Morphological and physiological traits were evaluated, including leaf symptoms, stem and root dry masses, leaf water potential, relative water content, chlorophyll content, photosystem II efficiency, hydrogen peroxide, thiobarbituric acid reactive substances, bioaccumulation and translocation factor estimates and removal efficiency for various heavy metals. Application of 25% IWW stress affected many aspects of plant morphology: chlorosis and necrosis in leaves, epinasty, leaf curling, early leaf senescence and root browning. In both species, the 25% IWW treatment reduced leaf, stem and root dry masses relative to controls. *S. alba* exhibited greater removal capacity for heavy metal ions and could be effective as a remediator of toxic-metal-polluted industrial effluent water.

**Keywords:** industrial wastewater; toxicity; heavy metals; *Salix alba*; *Casuarina glauca*; rhizofiltration





## 1. Introduction

Intensification of global industrialization has led to the release of large amounts of industrial wastewater (IWW). In North African countries, enormous volumes of IWW are dumped into the environment without sufficient treatment [1]. The main industrial users of water are electrical power stations, oil refineries, manufacturing, cooling tower usage and supplying boilers. Wastewater is discharged by industries following the extraction and smelting of metals, the production of beverages, textiles, paints, paper, plastic and tanneries, and the manufacture of pesticides and phosphate fertilizers [2]. It has been widely reported that industrial wastewater, especially discharges from the petrochemical industry, often contains high levels of the heavy metals Cd, Co, Fe, Mn, Ni and Zn, with concentrations of 0.015, 0.13, 2.98, 0.95, 1.57 and 8.07 mg $L^{-1}$, respectively, observed [3]. Along with heavy metals, IWW carries small quantities of various organic chemical contaminants, such as detergents and fossil fuels from the petroleum industry. Indeed, they are loaded with many constituents: oils, fats, suspended solids and organic matter of various origin, and they are often characterized by high levels of salts [4]. Heavy metals in IWW are important pollutants given their persistence in the environment and at high concentrations can induce



toxicity through the formation of free radicals that cause oxidative stress [5]. These heavy metals are by nature non-biodegradable and chemically non-oxidizable and tend to reduce soil quality by inhibiting the development of microflora and through their accumulation in superficial horizons, or they are transported in runoff to groundwater and surface water bodies, contaminating them [6]. Heavy metals can also be bioconcentrated with increasing trophic levels [6].

Faced with these negative effects of IWW and the increasing public awareness of environmental issues, it is essential to find solutions that would limit the risks associated with pollutants in wastewater. Different electrochemical, physical and chemical methods have been developed to treat IWW [7]. These methods are robust but face major disadvantages, such as low selectivity, limiting the removal of metals from industrial wastewater [8]. In fact, the overall goal of wastewater treatment is to treat it in the most cost-effective manner as possible in an environmentally friendly manner [9]. Indeed, scientists continually search for the best way to purify IWW at minimum expense and with maximum efficiency. Currently, biological techniques are strongly recommended as they are more in line with the principles of sustainable development. Rhizofiltration is an ecological and promising technology applicable to the remediation of contaminated water by using the plant root system to fix, extract, immobilize and adsorb toxic ions from soil or water solution [10]. This strategy combines phytoextraction and phytostabilization [10]. It is a relatively inexpensive phytotechnology, and serves as a clean and effective alternative solution for reducing water pollution [11]. Thus, phytoremediation methods seem to overcome heavy metal pollution even at low concentrations [12,13].

More than 450 plant species have been identified as being useful in rhizofiltration, ranging from floating aquatic plants such as *Eichhornia crassipes* [14], *Pistia stratiotes* [14], *Nuphar* and *Limnobium spongia* [15] to emergent aquatics such as *Scirpus*, *Phalaris arundinacea* and *Juncus* [15] to submerged aquatics such as *Elodea* [15]. This identification has continued with herbaceous terrestrial species that are grown in hydroponic systems, such as *Brassica juncea* [16], *Helianthus annuus* [17] and *Zea mays* [15]. These herbaceous plants are characterized by low biomass production [18]. In this study, interest is directed towards woody forest species that would facilitate accumulation of high levels of pollutants in their biomass [18]. Therefore, these species offer an opportunity for ecologically mediated disposal of wastewater as a source of irrigation and fertilization. This can be achieved via their ability to transpire large amounts of water, the production of large quantities of biomass, long lifespans, deeply penetrating root systems [18], rapid sap rise, rapid vegetative growth [19], ease of handling and harvesting [19] and their regrowth potential [19].

The use of poplars (*Populus*) and willows (*Salix*) has been shown to recycle water that is high in salinity and rich in boron [20] to treat larger amounts of dissolved pollutants including Cd and Zn, and to absorb excess nitrates and phosphates. For example, the spontaneous hybrid of *Salix viminalis* L. and *S. caprea* (*Salix × smithiana* Willd.) is characterized by a high capacity to accumulate Cd and Zn in the shoots [21]. *Paulownia* [22] and *Eucalyptus* [23] are promising genera, given the ability of their species to tolerate high concentrations of metals [24]. Seedling roots of *Casuarina glauca* Sieb ex Spreng can scavenge Cd, Pb, Ni and Zn ions at concentrations of 0.13, 1.31, 0.53 and 6.92 mg L$^{-1}$, and with removal efficiencies reaching 92%, 77%, 83% and 73%, respectively [13]. These plant species, especially their roots, show a wide range of tolerance to wastewater and heavy metal accumulation [13] via inhibiting the translocation of metals from roots to shoots (rhizoaccumulation), which is an essential process for phytostabilization purposes [25]. Nevertheless, it is widely accepted that rhizofiltration with hyperaccumulator woody species is a very interesting and feasible approach. These species can treat several types of contaminants, even at low concentrations [12].

Therefore, the present study was conducted to evaluate (*i*) the ecotoxicity of industrial wastewater and its effect on the ecophysiology of two tree species, i.e., *Casuarina glauca* and *Salix alba*, and (*ii*) to evaluate their rhizofiltration potential for possible applications in phytoremediation.

## 2. Materials and Methods

### 2.1. Plant Material

*Casuarina glauca* seedlings of the same size (mean $\pm$ SD: height, $80.0 \pm 3.0$ cm; root collar diameter, $5.0 \pm 1.2$ mm, n = 36) were produced from seeds at the forest nursery of El Agba, Manouba, Tunisia ($36°46'34''$ N, $8°41'05''$ E). *Salix alba* seedlings (height, $20.0 \pm 1.4$ cm; diameter, $10.8 \pm 1.4$ mm, n = 36) were produced from cuttings in Ain Drahem, Tunisia ($36°46'34''$ N, $8°41'05''$ E). Techniques and cultural practices for seedling and cutting production in modern and traditional forest nurseries are described in detail in our previous publications [26–28].

### 2.2. Experimental Design

The experimental design was installed at the National Institute for Research in Rural Engineering, Water and Forests (INRGREF), Tunisia ($36°50'$ N, $10°14'$ E, 3 m asl). *Casuarina glauca* and *Salix alba* seedlings were placed under hydroponic conditions in plastic pots (20 cm diameter $\times$ 17 cm deep; volume, 5.3 L) containing tap water. The experimental design consisted of six complete random blocks (four pots per block and twelve pots per species). In each pot, three seedlings of the same species were fixed in holes in the lids of the pots. Seedlings were kept suspended by strips of Styrofoam, while half of their length was subjected to external environmental conditions (photoperiod, 14 h; daily temperature, $29.6 \pm 4$ °C; relative humidity, $64 \pm 5\%$) and the other half was submerged. The medium was renewed three times each week. Both species were subjected to an acclimatization period of 3 months or 90 days (March–June 2021) to allow full development of leaves and roots. Five different concentrations of IWW (0, 25, 50, 75, 100% IWW) were used to evaluate the ecotoxicity of industrial wastewater and its effect on the ecophysiology of *Casuarina glauca* and *Salix alba*, but a concentration of IWW higher than 25% was found to be lethal to seedlings. Thus, only two concentrations (treatments) were tested: 0% IWW (control: C) and 25% IWW (stressed treatment: S). Six pots of each species remained filled with tap water, and the plants served as controls (control: C, tap water). For the other six pots of each species, tap water was replaced with 25% IWW, whereby the seedlings were stressed (stress: S, 25% IWW). A total of 72 seedlings were deployed as 3 seedlings $\times$ 2 species $\times$ 2 treatments $\times$ 6 blocks. The treatments and the species were randomly distributed within each block. Once the roots were saturated with heavy metals [29], the morphology and growth were affected; they were then harvested and removed [29]. The treatment lasted 35 days. The experiments were conducted in a greenhouse and the pots were continuously aerated using two types of oxygen pumps (one pump per four pots). In assessing effects of industrial wastewater on growth and the appearance of several symptoms in *Casuarina glauca* and *Salix alba* seedlings, visual checks of shoot and root morphology were made every three days.

### 2.3. Composition of Industrial Wastewater

The wastewater that was used in this study was collected from the Tunisian Electricity and Gas Company (STEG) in La Goulette, Tunisia ($36°49'09''$ N, $10°18'22''$ E). The initial physicochemical composition of industrial wastewater prior to dilution (100% IWW) is summarized in Table 1, which is compared with World Health Organization standards [30]. The physicochemical characterization of diluted industrial wastewater (25% IWW) was carried out on two dates: before purification (25% IWW) and at the end of the experiment after 35 days of purification (treated diluted industrial wastewater: T 25% IWW). T 25% IWW was collected in sterilized plastic bottles and transported to STEG for further laboratory analyses. Heavy metal concentrations were determined via flame atomic absorption spectrometry (ContrAA 300, ANALYTIK JENA, France). The determination of nitrite ($NO_2^-$), nitrate ($NO_3^-$) and suspended matter (SM) was conducted using visible spectrophotometry (DR 2800 model, HACH, Loveland, CO, USA). Electrical conductivity (EC, $\mu S\ cm^{-1}$) and the hydrogen ion potential (pH) were measured using a multiparameter probe (HQ30D model, HACH).

**Table 1.** Composition of initial industrial wastewater before dilution (100% IWW) and World Health Organization (WHO) standard. Three replications ± SD of each variable.

| Variable | 100% IWW | WHO Standard 2021 |
|---|---|---|
| pH $_{H_2O_2}$ | 7.46 ± 0.01 | 6.5–8.4 |
| CE (mS cm$^{-1}$) | 5.31 ± 0.02 * | 3 |
| MES (mg L$^{-1}$) | 200.0 ± 0.009 * | 100 |
| Si (mg L$^{-1}$) | 15.6 ± 0.13 | – |
| Ca (mg L$^{-1}$) | 660.0 ± 0.02 * | 200 |
| Mg (mg L$^{-1}$) | 240.0 ± 0.005 * | 150 |
| Zn (mg L$^{-1}$) | 1.56 ± 0.002 | 2 |
| Na (mg L$^{-1}$) | 2300.0 ± 0.9 * | 200 |
| Mn (mg L$^{-1}$) | 1.12 ± 0.09 * | 0.2 |
| K (mg L$^{-1}$) | 60.8 ± 0.04 | – |
| Co (mg L$^{-1}$) | 1.28 ± 0.11 * | 0.05 |
| Fe (mg L$^{-1}$) | 0.894 ± 0.017 * | 0.2 |
| Cu (mg L$^{-1}$) | 0.58 ± 0.007 | 2 |
| Ni (mg L$^{-1}$) | 0.008 ± 0.0005 | 0.07 |
| Cr (mg L$^{-1}$) | 0.012 ± 0.008 | 0.05 |
| Cd (mg L$^{-1}$) | 0.001 ± 0.0001 | 0.003 |
| Mo (mg L$^{-1}$) | 0.018 ± 0.0007 | 0.07 |
| NO$_2^-$ (mg L$^{-1}$) | 156.0 ± 0.005 * | 3 |
| NO$_3^-$ (mg L$^{-1}$) | 148.0 ± 0.002 * | 30 |
| C$_2$H$_6$O$_2$ (mg L$^{-1}$) | 128.0 ± 0.001 * | 26 |
| RD11 (mg L$^{-1}$) | 156.0 ± 0.005 * | 20 |

pH $_{H_2O_2}$ hydrogen ion potential of water, CE electrical conductivity, MES suspended matter, Si silicon, Ca calcium, Mg magnesium, Zn zinc, Na sodium, Mn manganese, K potassium, Co cobalt, Fe iron, Cu copper, Ni nickel, Cr chromium, Cd cadmium, Mo molybdenum, NO$_2^-$ nitrite, NO$_3^-$ nitrate, C$_2$H$_6$O$_2$ ethylene glycol. RD11 detergents used as antifreeze and anticorrosion to protect installations. * Higher than the limits of the World Health Organization standard of wastewater.

The efficiency of the 25% IWW treatment with *C. glauca* and *S. alba* seedlings was evaluated as described by Wang et al. [31] using the following formula:

$$E\ (\%) = (C_i - C_f/C_i) \times 100 \tag{1}$$

where E (%) is the efficiency of treatment, $C_i$ is the initial elemental concentration in the 25% IWW and $C_f$ is the final elemental concentration after 35 days in the T 25% IWW.

### 2.4. Determination of Leaf, Stem and Root Dry Masses

Dry mass was obtained after rinsing the roots with tap water and then reweighing the seedlings before and after 48 h at 70 °C. The seedlings were separated into stems, leaves and roots. Dry mass (g DM/seedling) was recorded using a precision balance with three decimal places (0.001g).

### 2.5. Determination of Chlorophyll Concentration (SPAD Values) and Fluorescence

Chlorophyll concentrations were measured on the leaves (or needles) that had been most exposed to natural environmental conditions (photoperiod, 14 h; daily temperature, 29.6 ± 4 °C; relative humidity, 64 ± 5%) with a chlorophyll meter (SPAD-502 model, Konica Minolta, Tokyo, Japan) with a measurement accuracy within ±1.0 SPAD unit [32]. SPAD-502 m provides an alternative means of measuring relative leaf chlorophyll levels that overcomes the drawbacks of the organic solvent pigment extraction method [33]. Indeed, SPAD values (generally between 0.0 and 50.0) correlate strongly with direct photometric measurements of chlorophyll (nmol cm$^{-2}$) that is extracted from the leaf [33]. Five randomly selected leaves (or needles of each twig) were used from each treatment combination (control: C, 0% IWW vs. stress: S, 25% IWW) per block, with five replicates per measurement.

Initial fluorescence ($F_0$), maximum fluorescence ($F_m$), variable fluorescence ($F_v = F_m - F_0$) and maximum quantum yield of PSII ($F_v/F_m$) were measured using a portable infrared fluorometer (LCpro-SD, ADC Bioscientific Ltd., Hoddesdon, Herts, UK). Five randomly selected leaves (or needles of each twig) from each seedling/species/treatment (C and S)/block were used, with five replicates per measurement. All measurements were taken on leaves (or needles) after dark adaptation overnight. The potential quantum yield of photosystem II (PSII), which was expressed as $F_v/F_m$, was calculated as follows [34]:

$$F_v/F_m = (F_m - F_0)/F_m \tag{2}$$

Chlorophyll concentrations (SPAD values) and fluorescence were measured on the middle part of leaves (or needles) and on the uppermost portion of seedlings that were fully developed [35], after 12 days from the start of the experiment.

### 2.6. Determination of Leaf Water Potential and Relative Water Content

Leaf water potential was measured via the Scholander pressure chamber technique (600 model, PMS Instrument Company, Albany, OR, USA) on young, fully developed leaves. The freshly harvested leaf (or needle) was inserted into a gas-tight stopper with the cut end protruding a few millimeters from the stopper. Once sealed in the chamber, pressure was applied and monitored until sap was expressed from the cut stem's surface.

Relative water content (RWC) was determined according to the protocol described by Scotti-Campos et al. [36]. Six randomly selected leaves (or needles) from each seedling/species/treatment (C and S)/block were used. The leaves (or needles) were first weighed to determine the fresh weight (FW). They were then cut at the petiole and floated in test tubes containing 15 mL of distilled water for 24 h at ambient temperature in the dark. The samples were dried on filter paper and reweighed to determine the turgor weight (TW). They were then oven-dried (80 °C for 48 h) and reweighed to obtain the dry weight (DW). RWC was calculated according to the following formula [37]:

$$RWC\ (\%) = [(FW - DW)/(TW - DW)] \times 100 \tag{3}$$

### 2.7. Determination of the Integrity of Membrane Structures, Hydrogen Peroxide ($H_2O_2$) and Thiobarbituric Acid Reactive Substances (TBARSs)

Membrane permeability was measured by determining electrolyte leakage induced by metallic stress. Electrolyte leakage was measured according to the protocol described by Thiaw [38]. One gram of fresh leaf material/species/treatment was cut into pieces 1 cm in length and rinsed three times with distilled water (DW) in Petri dishes, and then floated in glass tubes containing 15 mL of DW in order to eliminate the electrolytes released following excision. The samples were then placed in a water bath at 40 °C in the dark for 1 h and cooled to room temperature to measure the free conductivity (FC) of the supernatant using a conductivity meter (Cellox 325 model, Multiline P3 PH/LF-SET, WTW GmbH, Weilheim, Germany) and expressed in $\mu S\ cm^{-1}$. The value that was obtained corresponds to the residual tonoplast permeability to ions. After this measurement, these samples that had been soaked in water were placed in a water bath boiling at 100 °C for 20 min to destroy the leaf tissue. After cooling to ambient temperature, total conductivity (TC) was measured. The test was repeated five times; we determined the percentage of damage (PD%) for each treatment (C, S) according to the following formula:

$$PD\ (\%) = 100 - PRI \tag{4}$$

$$PRI\ (\%) = (PAI\ stress/PAI\ control) \times 100 \tag{5}$$

$$PAI\ (\%) = (1 - FC/TC) \times 100 \tag{6}$$

where PRI is the percentage of relative integrity and PAI is the percentage of absolute integrity.

Hydrogen peroxide ($H_2O_2$) determination was carried out according to the method described by Sergiev et al. [39] using 20 randomly selected seedlings (5 leaves or needles/bloc/treatment (C and S)/species). To do this, 0.5 g of fresh material was homogenized with 5 mL of TCA (0.1%) with a cold (4 °C) mortar and pestle (i.e., 10 mL/g of fresh material). The homogenate was centrifuged at $12,000 \times g$ for 15 min at 4 °C. A 0.5 mL aliquot of the supernatant was mixed with 0.5 mL of phosphate buffer (10 mM, pH = 7) and 1mL of potassium iodide (KI) (1 M). Under the same conditions, a range of suitable concentrations of hydrogen peroxide were used for calibration. Absorbance was determined at 390 nm using a UV spectrophotometer (UV-3100 Model, Xian Yima Optoelec Co., Ltd., Xi'An, Shaanxi, China). The concentration of $H_2O_2$ was calculated using its molar extinction coefficient ($\varepsilon$ = 39.4 mM$^{-1}$ cm$^{-1}$).

Lipid peroxidation was performed by determining the concentration of thiobarbituric acid reactive substances (TBARSs) using the method of Heath et al. [40]. Twenty seedlings (5 leaves or needles/bloc/treatment (C and S)/species) were selected, and 1 g was ground in 10 mL of extraction buffer consisting of thiobarbituric acid (TBA: 0.5%) (*w/v*), trichloroacetic acid (TCA: 10%) (*w/v*) and 0.2 mM EDTA. The ground material was heated in a water bath at 95 °C for 30 min. The reaction between TBA and endoperoxides, primarily MDA (malondialdehyde), leads to the formation of a TBA-MDA complex [41]. The enzymatic reaction was stopped by immediate cooling in an ice bath at 4 °C. After centrifugation at 1000 rpm for 10 min, absorbance of the supernatants was measured at 532 nm and 600 nm against a blank using a UV spectrophotometer (UV-1200 Model, Tomos Life Science Group Pte, Ltd., China). TBARS concentrations were calculated using the extinction coefficient ($\varepsilon$ = 155 mM$^{-1}$ cm$^{-1}$).

### 2.8. Potential for Rhizofiltration

At the end of the experiment, the shoot and root parts of the seedlings of *C. glauca* and *S. alba* were harvested, rinsed with distilled water, dried separately at 105 °C for 72 h and ground. For each species, five seedlings per treatment (C and S) were randomly selected, after which 0.2 g samples were transferred into numbered Teflon tubes to be digested in $HNO_3$-$H_2SO_4$-$HClO_4$ (4:4:2) for 2 h at 300 °C. After digestion and mineralization, the mixture was filtered, and then the volume was adjusted to 100 mL with distilled water. Quantification of metallic trace elements was conducted using a flame mode atomic absorption spectrometer (ContrAA 300, ANALYTIK JENA, France). Nitrite and nitrate assays were carried out with a visible spectrophotometer (DR 2800, HACH). Results are expressed in mg g$^{-1}$ DM.

The phytoremediation potential of *Salix alba* and *Casuarina glauca* was assessed at the end of the experiment, using bioaccumulation (BAF) and translocation (TF) factors [42]:

$$BAF = \text{ion concentration in the seedling/ion concentration in 25\% IWW} \qquad (7)$$

$$TF = \text{ion concentration in shoots/ion concentration in roots} \qquad (8)$$

### 2.9. Statistical Analyses

Statistical analyses were performed with SPSS 22.0 software (IBM, Armonk, NY, USA). One-way ANOVA [43] was conducted on each response variable. Means comparisons of the measurements (dry mass seedling growth, water status, chlorophyll concentration and fluorescence, TBARSs, $H_2O_2$ and integrity of membrane structures) relating to the two treatments (control C vs. stress S) in the two species (*S. alba* and *C. glauca*) was carried out with Student–Newnan–Keuls (SNK) tests at a significance level of 5%. Each value is presented as the mean $\pm$ standard deviation (SD).

## 3. Results

### 3.1. Physicochemical Composition of Initial Industrial Wastewater before Dilution (100% IWW)

The compositions of 100% IWW and World Health Organization (WHO) standard values [30] are summarized in Table 1. This 100% IWW exhibited a high electrical conductivity and a high load of suspended matter relative to the standard reference values demanded by the WHO (Table 1). Concentrations of Ca, Mg, Na, Mn, Fe, Co, $NO_2^-$ and $NO_3^-$ were elevated in 100% IWW (Table 1). The 100% IWW included two types of detergents, ethylene glycol and RD11, whose concentrations were almost 5-fold and 8-fold higher, respectively, than the standard WHO values (Table 1).

### 3.2. Morphological Aspects and Growth of Seedlings

For *Salix alba* seedlings that were treated with 25% IWW (stressed treatment: S), the appearance of chlorosis and progressive necrotic patches on the edges of the leaf blades were recorded after 6 days (Figure 1B). In addition, preferential downward inclination of the leaves (=epinasty) (Figure 1D) that was accompanied by leaf rolling was observed after six days (Figure 1C). For *Casuarina glauca* seedlings that were treated with 25% IWW (stressed treatment: S), progressive drying of the needles was recorded beyond day 15 (Figure 1F).

The stressed treatment (S) caused browning of the initial root system in the two species from day 3 onward. After 12 days, we noted the emergence of new white roots in seedlings of *S. alba* treated with 25% IWW (Figure 2C), followed by budding of the buds at about day 15 (Figure 2A). Additionally, beyond 21 days, the emergence of new leaves of a smaller size was noted in *Salix alba* that was treated with 25% IWW (S) (Figure 2B). At the end of the treatments, the stressed treatment that was imposed on the seedlings of *S. alba* (S) and *C. glauca* (S) caused a significant loss of leaf dry mass (Figure 3A,D; $p < 0.001$, $p < 0.017$), stem dry mass (Figure 3B,E; $p < 0.043$, $p < 0.051$) and root dry mass (Figure 3C,F; $p < 0.01$, $p < 0.297$) compared to controls. Subsequent reductions in the dry mass of leaves, stems and roots compared to the control were 58, 37.7 and 74.6% in *S. alba* (S), yet only 34.6, 28.3 and 24.4%, respectively, in *Casuarina glauca* (S) (Figure 3A–F).

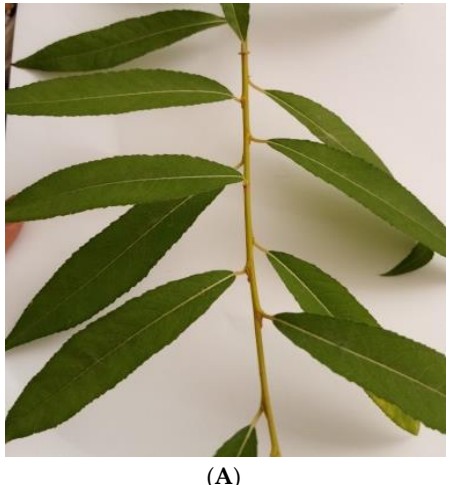 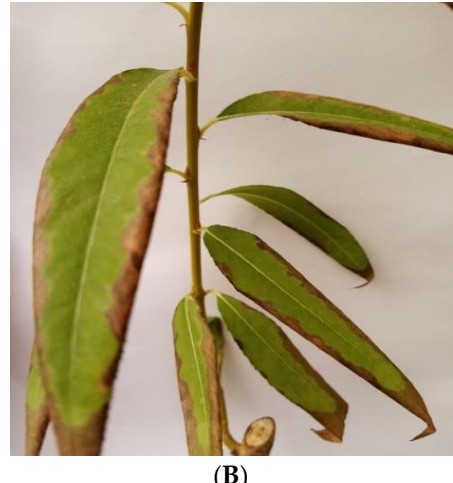

(**A**)　　　　　　　　　　　　　　　　　(**B**)

**Figure 1.** *Cont.*

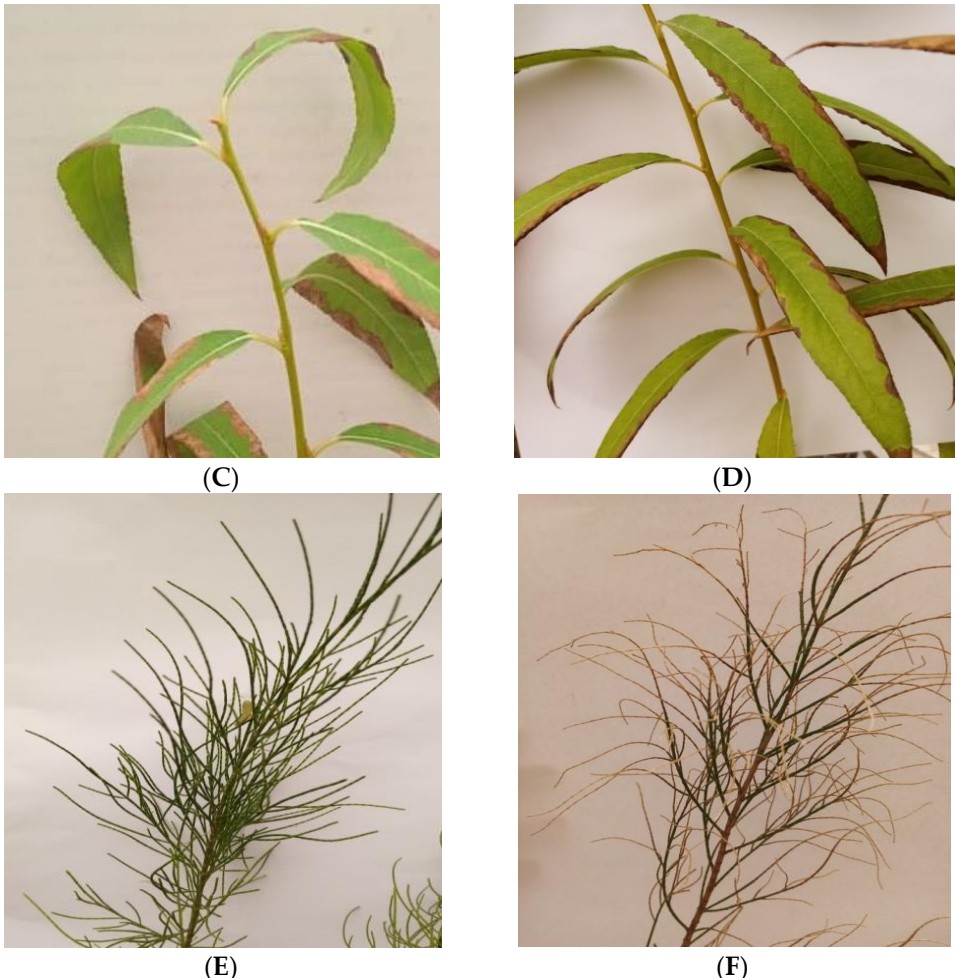

**Figure 1.** Appearance of chlorosis and necrotic areas on the edges of the limbus (**B**), leaf rolling (**C**) and epinasty (**D**) after 6 days in seedlings of *S. alba* treated with 25% industrial wastewater (25% IWW) compared to control (**A**), and desiccation of the needles of *C. glauca* treated with 25% IWW after 15 days (**F**) compared to control (**E**).

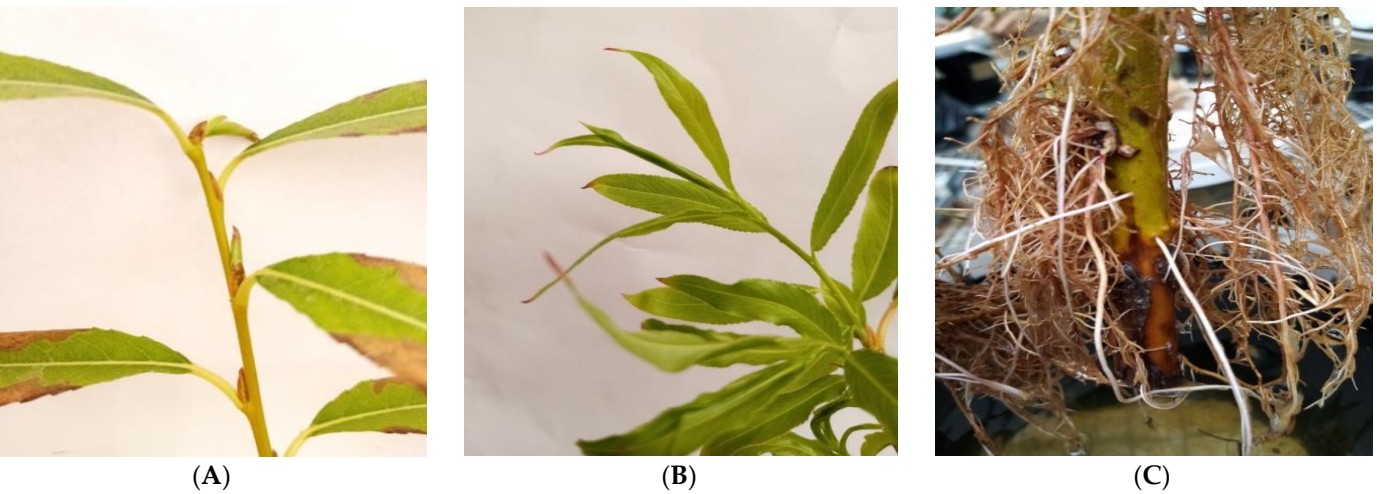

**Figure 2.** The appearance of a recovery phase in *S. alba* treated with 25% of industrial wastewater (25% IWW), which manifested as bud burst after 15 days (**A**), emergence of new leaves after 21 days (**B**) and formation of white roots after 12 days (**C**).

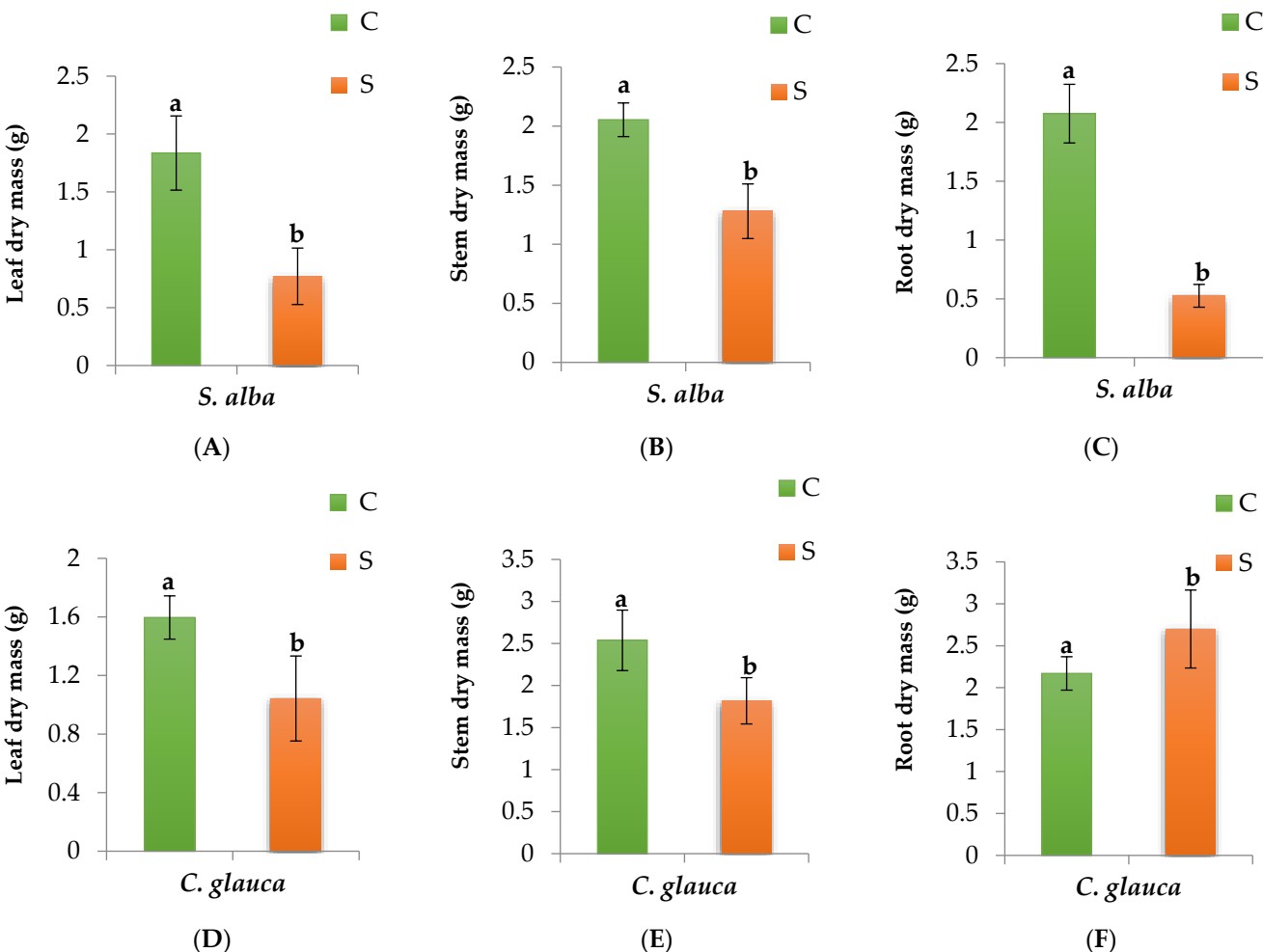

**Figure 3.** Comparison of leaf (**A**,**D**), stem (**B**,**E**) and root dry masses (**C**,**F**) in seedlings of *S. alba* and *C. glauca* treated with tap water (control, C) or with 25% industrial wastewater (25% IWW; stressed, S) after 35 days of treatment, (n = 5, mean ± SD). Different letters above the means indicate a significant difference at $p < 0.05$ according to Student–Newnan–Keuls (SNK) tests.

### 3.3. Leaf Water Potential and Relative Water Content

Tissue relative water content (RWC) was significantly reduced during the treatment of the two species, *S. alba* (S) ($p < 0.045$) and *C. glauca* (S) ($p < 0.027$) (Figure 4C). After 30 days, these reductions were 12% and 16% in *S. alba* (S) and *C. glauca* (S), respectively (Figure 4C). Treatment with 25% industrial wastewater (25% IWW) (S) that was imposed on the seedlings of *S. alba* (S) and *C. glauca* (S) significantly lowered leaf water potential (Figure 4A). After 30 days, these values reached 0.6 and 1.21 MPa in *S. alba* (S) ($p < 0.005$) and *C. glauca* (S) ($p < 0.0001$), respectively (Figure 4A).

### 3.4. Chlorophyll Fluorescence and the Concentration of Chlorophyll (SPAD Value)

The (Fm-Fo)/Fm value reflects the photosynthetic efficiency of PS II in using light for photochemical conversion. Application of 25% industrial wastewater (25% IWW) significantly reduced chlorophyll fluorescence in *C. glauca* (S) ($p < 0.0001$), reaching 21% that of the control (Figure 4D). Moreover, the application of this treatment for 12 days caused a significant degradation of chlorophyll, which is explained by the decrease in SPAD values in *S. alba* (S) ($p < 0.0001$) and in *C. glauca* (S) ($p < 0.0001$) as compared to the control (Figure 4B). This reduction was 56% and 42%, respectively (Figure 4B).

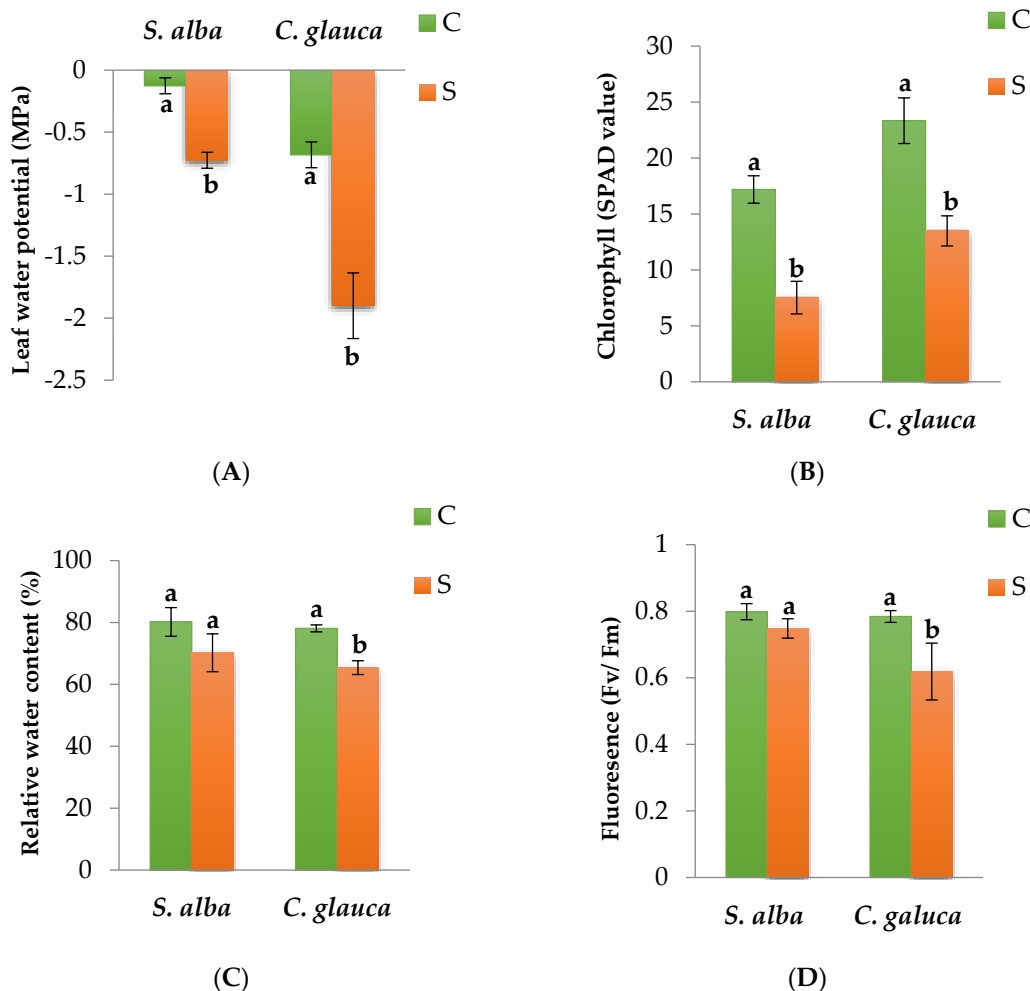

**Figure 4.** Variation in leaf water potential (**A**), SPAD value (**B**), relative water content (**C**) and chlorophyll fluorescence (**D**) measured after 12 days in *S. alba* and *C. glauca* treated with 25% industrial wastewater (25% IWW; stress, S) or with tap water (control, C), (n = 5, mean ± SD). Different letters above the means indicate a significant difference at $p < 0.05$ according to SNK tests.

### 3.5. Concentration of Thiobarbituric Acid Reactive Substances (TBARSs), Hydrogen Peroxide $H_2O_2$ and Percentage of Damage to Membrane Structures

The stress treatment (25% IWW) significantly increased *TBARS* concentrations in *S. alba* (S) ($p < 0.042$) and in *C. glauca* (S) ($p < 0.006$) (Figure 5B). These rates of increase were similar and averaged 50%. These results show that stress induced a significant increase in $H_2O_2$ levels in *S. alba* (S) ($p < 0.0001$) and in *C. glauca* (S) ($p < 0.0001$) (Figure 5A). The rates of increase were 1000% and 130%, respectively. The stress treatment significantly affected the stability of membrane structures in both species (Figure 5C). The percentage of damage to the membrane structures compared to the control was 15% and 65% in *S. alba* (S) and *C. glauca* (S), respectively (Figure 5C).

### 3.6. Accumulation and Compartimentation

Accumulation of ions (Na, Fe, Co and Mn) varied according to the species and the component being considered (shoot or root), as indicated in Table 2. In stressed seedlings of *Salix alba* (S), compared to the control, Fe and Mn levels decreased in leaves and increased in roots, while the opposite trend was observed for Na. With respect to Co concentrations, an increase was recorded in the shoots and roots of stressed seedlings (Table 2). In *C. glauca* (S), the accumulation of Co ions was greater in shoots than that observed for the roots, while the opposite was true for Na, Fe and Mn (Table 2). It should be noted that in *S. alba* (S), the ions that were measured (Na, Mn, Co and Fe) in old senescent leaves showed clear

accumulation, reaching concentrations of 0.4, 0.015, 0.03 and 0.08 mg g$^{-1}$, respectively (Table 2).

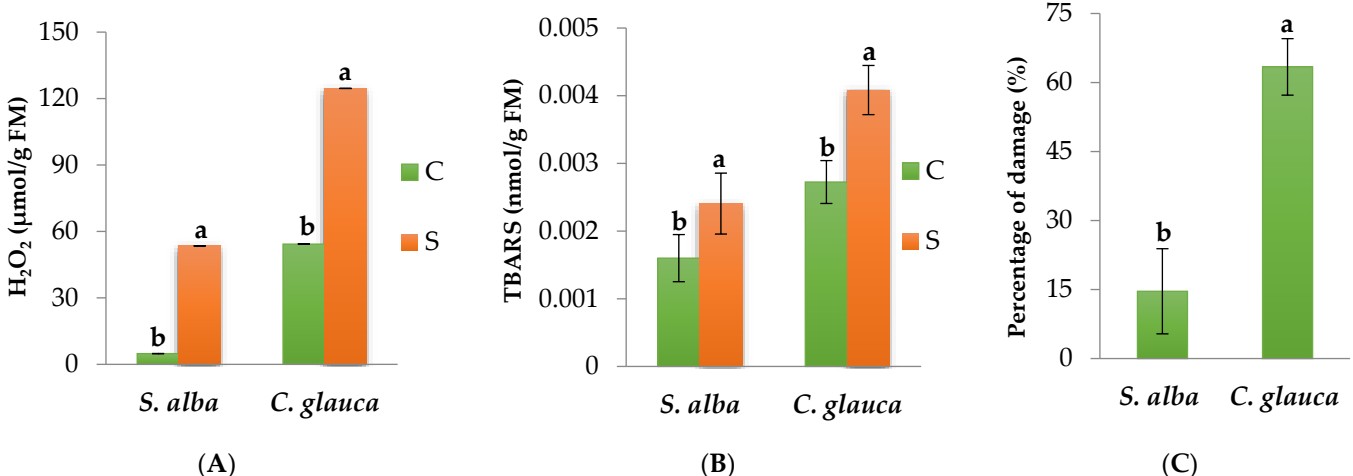

**Figure 5.** Variation in the concentrations of hydrogen peroxide H$_2$O$_2$ (**A**), *thiobarbituric acid reactive substances (TBARSs)* (**B**) and percentage damage to membrane structures (**C**) in *S. alba* and *C. glauca* treated with IWW (S, stress) or with tap water (C, control), (n = 5; mean ± SD). Different letters above the means indicate a significant difference at $p < 0.05$ according to SNK tests.

For the bioaccumulation (BAF) and translocation (TF) factors (Table 3) of the toxic ions, the BAF followed the order Na < Mn < Co < Fe and Na < Co < Mn < Fe for *S. alba* (S) and *C. glauca* (S), respectively (Table 3).The BAF was greater than 1 for Fe in *C. glauca* (S) and the young leaves of *S. alba* (S) and for Co only in the young leaves of *S. alba* (S) (Table 3). The TF is an indicator that aids in understanding the mobility of ions in the seedling. The TF followed the order Fe < Na < Mn < Co in newly formed *S. alba* (S) leaves, with the TF being greater than 1 for Co (Table 3). At the level of senescent leaves in *S. alba* (S), the TF followed the order Fe < Na < Co < Mn, where all values were greater than 1 (Table 3). In *C. glauca* (S), the TF followed the order Fe < Mn < Na < Co (Table 3).

**Table 2.** Variation in Na, Mn, Co and Fe ion concentrations in the shoot and roots of *S. alba* and *C. glauca* seedlings subjected to stress for 35 days imposed by 25% of industrial wastewater (S, stressed treatment) or in optimal conditions with tap water (C, control). Three replications ± SD of each parameter were carried out.

| | | | Ions (mg g$^{-1}$ DM) | | | |
|---|---|---|---|---|---|---|
| | | | **Na** | **Mn** | **Co** | **Fe** |
| *S. alba* | Root | C | 0.47 ± 0.002 a | 0.006 ± 0.0003 a | 0.007 ± 0.0002 a | 0.03 ± 0.008 a |
| | | S | 0.44 ± 0.002 a | 0.008 ± 0.0003 b | 0.02 ± 0.001 b | 0.12 ± 0.002 b |
| | Shoot | C | 0.15 ±0.001 a | 0.01 ± 0.005 a | 0.008 ± 0.0003 a | 0.04 ± 0.005 a |
| | | S | 0.18 ± 0.001 b | 0.006 ±0.005 a | 0.02 ± 0.003 b | 0.03 ± 0.005 a |
| *C. glauca* | Root | C | 0.62 ± 0.003 a | 0.008 ± 0.0007 a | 0.002 ± 0.0008 a | 0.07 ± 0.003 a |
| | | S | 0.67 ± 0.003 b | 0.01 ± 0.003 b | 0.005 ± 0.0006 b | 0.18 ± 0.002 b |
| | Shoot | C | 0.54 ± 0.001 a | 0.005 ± 0.0003 a | 0.008 ± 0.0004 a | 0.08 ± 0.005 a |
| | | S | 0.56 ± 0.001 a | 0.008 ± 0.0003 b | 0.009 ± 0.0003 b | 0.05 ± 0.002 a |

Na sodium, Mn manganese, Co cobalt, Fe iron. Note: for each element, means followed by distinct letters within the root or aerial parts show significant differences at the 5% level according to SNK tests.

**Table 3.** Bioaccumulation factor (BAF) and translocation factor (TF) in seedlings of *C. glauca* (S, stress) and *S. alba* (S, stress) treated with 25% industrial wastewater (S, stressed treatment) over 35 days.

| Ions (mg g$^{-1}$ DM) | S. alba (S) | | | | C. glauca (S) | |
|---|---|---|---|---|---|---|
| | Young Leaves | | Senescent Leaves | | | |
| | TF | BAF | TF | BAF | TF | BAF |
| Na | 0.41 | 0.001 | 2.26 | 0.0007 | 0.84 | 0.002 |
| Mn | 0.75 | 0.7 | 5 | 0.05 | 0.8 | 0.11 |
| Co | 1.15 | 1.12 | 4 | 0.13 | 0.86 | 0.14 |
| Fe | 0.2 | 1.02 | 1.1 | 0.4 | 0.3 | 1.05 |

Na sodium, Mn manganese, Co cobalt, Fe iron. Note: early senescence was only recorded in *S.alba* seedlings (S).

### 3.7. Rhizofiltration Potential

Physicochemical and heavy metal data of diluted industrial wastewater (25% IWW) and treated diluted industrial wastewater (T 25% IWW) are summarized in Table 4. The pH indicated a strong reduction, reaching 23.9% for *C. glauca* (S), while a low increase of 6.5% was attained for *S. alba (S)* (Table 4). Electrical conductivity (EC) was 44.8% and 42.1% lower after T 25% IWW treatment for *S. alba* (S) and *C. glauca* (S), respectively, as compared to the EC of 25% IWW treated plants (Table 4). Efficiency of treatment (E) significantly increased in *S. alba* (S) seedlings, compared to *C. glauca* (S), for Ca, Mg, Na, K, NO$_2^-$, NO$_3^-$, Mn, Co, Fe, Cu, Ni, Cr, Cd and Mo (Table 4). All ion concentrations were significantly reduced in T 25% IWW, indicating the efficiency of these two forest species, which are used to reduce or remove toxic ions from 25% IWW (Table 4).

**Table 4.** Physicochemical parameters (mean ± SD, n = 3) of diluted untreated industrial wastewater (25% IWW) over time: d = 0 day, of the diluted treated industrial wastewater (T 25% IWW) with *S. alba* or *C. glauca* after 35 days of treatment and the efficiency of this treatment (E) for these 2 species. Three replications ± SD of each parameter were carried out.

| Elements | 25% IWW (d = 0 day) | T 25% IWW + S. alba, T 25% IWW + C. glauca (d = 35 days) | | | |
|---|---|---|---|---|---|
| | | T 25% IWW | E (%) | T 25% IWW | E (%) |
| pH$_{H_2O_2}$ | 7.46 ± 0.01 | 7.95 ± 0.01 | – | 5.67 ± 0.01 | – |
| CE (mS cm$^{-1}$) | 5.31 ± 0.02 * | 2.93 ± 0.02 | – | 3.07 ± 0.02 | – |
| Ca (mg L$^{-1}$) | 165.0 ± 0.02 | 116.0 ± 0.02 | 29.6 | 133.5 ± 0.02 | 19 |
| Mg (mg L$^{-1}$) | 60.0 ± 0.005 | 38.5 ± 0.005 | 35.8 | 42.0 ± 0.005 | 30 |
| Zn (mg L$^{-1}$) | 0.389 ± 0.002 | <LQ | 100 | 0.16 ± 0.002 | 58.8 |
| Na (mg L$^{-1}$) | 575.0 ± 0.9 * | 460.0 ± 0.9 * | 20 | 495.0 ± 0.9 * | 13.9 |
| Mn (mg L$^{-1}$) | 0.28 ± 0.09 * | 0.025 ± 0.003 | 89.3 | 0.155 ± 0.030 | 42.9 |
| K (mg L$^{-1}$) | 15.2 ± 0.04 | 12.8 ± 0.04 | 15.7 | 12.7 ± 0.04 | 16.4 |
| Co (mg L$^{-1}$) | 0.32 ± 0.11 * | 0.042 ± 0.010 | 87.5 | 0.1 ± 0.01 * | 68.8 |
| Fe (mg L$^{-1}$) | 0.223 ± 0.017 * | 0.149 ± 0.017 | 31.8 | 0.168 ± 0.017 | 22.7 |
| Cu (mg L$^{-1}$) | 0.144 ± 0.007 | <LQ | 100 | <LQ | 100 |
| Ni (mg L$^{-1}$) | 0.002 ± 0.0005 | <LQ | 100 | <LQ | 100 |
| Cr (mg L$^{-1}$) | 0.002 ± 0.0008 | <LQ | 100 | <LQ | 100 |
| Cd (mg L$^{-1}$) | 0.0004 ± 0.0001 | <LQ | 100 | <LQ | 100 |
| Mo (mg L$^{-1}$) | 0.005 ± 0.0007 | <LQ | 100 | <LQ | 100 |
| NO$_2^-$ (mg L$^{-1}$) | 39.0 ± 0.005 * | 5.8 ± 0.005 * | 85.1 | 6.1 ± 0.005 * | 84.3 |
| NO$_3^-$ (mg L$^{-1}$) | 37.0 ± 0.002 * | 4.2 ± 0.002 | 88.6 | 5.9 ± 0.002 | 84.3 |

pH $_{H_2O_2}$ hydrogen potential of water, CE electrical conductivity, Ca calcium, Mg magnesium, Zn zinc, Na sodium, Mn manganese, K potassium, Co cobalt, Fe iron, Cu copper, Ni nickel, Cr chromium, Cd cadmium, Mo molybdenum, NO$_2^-$ nitrite, NO$_3^-$ nitrate. LQ limit of quantification * Higher than the limits of the World Health Organization standard.

## 4. Discussion

### 4.1. Morphological Changes and Seedling Growth

The industrial wastewater (IWW) applied at 25% IWW to *S. alba* (S, stress) and *C. glauca* seedlings (S) caused the onset of several symptoms related to the toxicity of metal ions (pronounced chlorosis, progressive necrosis, early senescence and root browning) (Figure 1B–D,F). This adverse effect of heavy metals appears at the whole-seedling level and was attributed to excessive accumulation of Mn, Co, Fe and Na (Table 2), which is associated with physiological disturbances [44] and nutritional imbalance [45]. In addition, leaf rolling in *S. alba* (S) decreases leaf area and can lead to a decrease in photosynthetic activity [46]. Similar signs were observed in *S. viminalis* [47]. Moreover, exposure to 25% IWW affected root morphology and caused progressive root browning, leading to a reduction in water and mineral uptake by the roots. Such morphological disturbances could be attributed to a calcium deficiency [48] or to an accumulation of potentially toxic elements, thereby limiting their transport to the leaves [49]. This highlights the premature senescence of older leaves and the appearance of a recovery phase with the emergence of new leaves and white roots, markedly so in *S. alba* (S) (Figure 2A–C). This response has been reported to be an adaptive strategy following vacuolar sequestration of the most toxic heavy metals [50].

The 25% IWW caused a significant reduction in the growth of *S. alba* (S) and *C. glauca* seedlings (S) (Figure 3A–F). Similar results were observed with *Populus* sp. [51]. In *S. alba* (S) that was treated with 25% IWW, heavy metals were preferentially accumulated in older leaves during the process of leaf senescence (Figure 3A). The decrease in whole-plant dry mass could be a direct consequence of excessive accumulation and effects of heavy metal ions in the cells [52,53], and can also occur with higher levels of Na. Our results (Table 4) are consistent with other studies showing the stress-reducing effects of excess Na on plant growth through negative impacts on net assimilation in *Salix viminalis* plants that were irrigated with wastewater containing high Na concentrations [45].

### 4.2. Water Status, Chlorophyll Concentration and Chlorophyll Fluorescence

The significant decreases in relative water content (RWC) and water potential in both species (Figure 4A,C) testify to the severity of the stress that was imposed by 25% IWW. In effect, osmotic stress incurred by the excess of ions would result in a reduction in the supply of water to the leaf tissues. Beyer et al. [43] showed that excess metallic elements limit absorption of essential ions and can cause a hydromineral imbalance.

This growth retardation in seedlings could be explained by a dysfunction of the photosynthetic system due to an impoverishment (depletion) of chlorophyll pigments (Figure 4B). This reduction has been attributed to an inhibition of the chlorophyll biosynthetic pathway [54], as seen in *Populus* [55], and the formation of a proteolytic enzyme responsible for chloroplast degradation [54]. Regardless, under the effect of 25% IWW treatment, degradation of chlorophyll pigments (Figure 4B) could be attributed to a destruction of the chloroplast membranes, an increase in the activity of chlorophyllase or a reduction in the absorption of Mg ions [56] or Fe, which is an essential cofactor for the synthesis of chlorophyll [57]. In *S. alba* (S) and *C. glauca* (S), this decline in chlorophyll (Figure 4B) led to a decrease in the functioning of the photosynthetic system II (PSII) (Figure 4D). This loss of PS II effectiveness would generally result in stomatal closure [58], which is often considered an indicator of root stress [59]. It should be noted that the leaf rolling recorded in *S. alba* (S) (Figure 1C) contributes to limiting light interception and would result in a reduction in transpiration [60]. It was shown that irrigation with wastewater caused a significant reduction in photosynthetic characteristics in two cultivars of *Triticum aestivum* L. (wheat), i.e., Chamran and Behrang, most notably in terms of chlorophyll fluorescence (Fv/Fm) and photosynthetic pigments [3].

### 4.3. Traits Promoting Metal Stress Tolerance

Our results show that 25% IWW toxicity caused significant tissue accumulation of hydrogen peroxide ($H_2O_2$) (Figure 5A). Heavy metals induce the overproduction of reactive oxygen species (ROS), in particular $H_2O_2$ [61], which are likely to cause membrane disruption [12] and an alteration in the photosynthetic electron transport chain [62]. Other observations showed that high $H_2O_2$ concentrations induced severe disturbances at the cellular level and generally translate into electrolyte leakage [63]. Furthermore, $H_2O_2$ is a cellular response that signals the presence of stress [64]. At low concentrations, $H_2O_2$ can function as a secondary messenger that activates antioxidant defenses [65], as is the case in *S. alba* seedlings (S) (Figure 5A). This defensive behavior in *S. alba* (S) resulted in greater stability of membrane structures (Figure 5C). Membranes are selective barriers that control the diffusion of ions. They must overcome the harmful effects of heavy metals [66], generally leading to the loss of membrane integrity [67]. Our results show a significant increase in thiobarbituric acid reactive substances (TBARSs) in both species (Figure 5B), i.e., a cytotoxic product of membrane lipid peroxidation [68].

### 4.4. Metal Bioaccumulation and Rhizofiltration Potential

For dosed ions (Na, Mn, Co, Fe), seedlings of both species showed variability in extraction capacity (Table 3). This behavior is correlated with a strategy for excluding heavy metals opted by seedlings that minimizes their harmful effects [69]. Weak translocation of heavy metals towards aerial parts is considered as a defense mechanism in seedlings to protect the photosynthetic apparatus [70]. About 75 to 90% of the heavy metals that are adsorbed by the seedling are blocked at the level of the roots [71]. This limitation of the mobility of heavy metals results from their complexation, transport and sequestration in root vacuoles [72]. Thus, the roots act as a trap organ that reduces transfer of heavy metals to aerial tissues, thereby limiting their toxicity and improving tolerance in the plant [73]. This has been considered as a resistance mechanism where an heavy metals excluding capacity is manifested in the face of high external heavy metal concentrations [73]. In stressed plants, the decrease in the concentrations of certain nutrients can be attributed to antagonistic interactions between nutrients and heavy metal exclusion [73]. As was seen in the work of Hajihashemi et al. [3], toxic levels of mineral elements in wastewater resulted in a significant drop in the content of K and Zn from the leaves in two cultivars of Triticum aestivum L. (wheat), i.e., Chamran and Behrang. The values of translocation factor TF < 1 in *C. glauca* (S) and *S. alba* (S) for Na, Mn and Fe testify to their aptitude for phytostabilization (Table 3). Comparable results have been reported by many authors [54,74]. In addition, other species limit the absorption of heavy metals in the roots due to avoidance or tolerance strategies [75]. Plants reorganize their root architecture to avoid growth in contaminated soils [76] that would maintain low ion concentrations in aerial parts [77]. Our results also showed that for senescent leaves of *S. alba* (S), TF values were >1 for all toxic ions. All concentrations of toxic ions were significantly reduced in treated diluted industrial wastewater (T 25% IWW) with *S. alba* (S) and *C. glauca* (S) compared to the 25% IWW, confirming the effectiveness of the two species in the removal of toxic ions (Table 4). Slaimi et al. [13] showed similar results with *C. glauca* seedlings (S). The preferential order of toxic ion accumulation in *C. glauca* (S) was noted as Na < Mn < Co < Fe and in *S. alba* as Na < Co < Mn < Fe. The bioaccumulation factor (BAF) was always < 1 for both species, except in *C. glauca* (S) for the Fe ion (Table 3). This high BAF value confirmed that *C. glauca* (S) could be a suitable seedling species for Fe phytoextraction as accumulator seedlings showed BAF values > 1 [13]. The BAF is an index of a plant's ability to accumulate a toxic ion depending on its concentration in the medium [78]. The high values recorded in this experiment are confirmed by Zacchini et al. [79].

The high removal efficiency (E), ranging from about 13% to 99%, which was observed in both species, testifies to their capability to extract toxic ions from 25% IWW (Table 4). Several studies have reported the use of rhizofiltration systems for the purification of water contaminated with heavy metals [80].

## 5. Conclusions and Research Needs

In the present study, two woody forest species, *S. alba* and *C. glauca*, were explored for the purification via rhizofiltration of diluted industrial wastewater (25% IWW) contaminated with heavy metals. This treated diluted industrial wastewater (T 25% IWW) can be used as a source of irrigation and fertilization. Heavy metals impose negative effects on seedling growth by decreasing leaf water potential, inducing morphological disturbances (chlorosis, necrosis, epinasty, leaf rolling and early senescence) and changes in physiological processes (photosynthetic dysfunction activity, a reduction in chlorophyll fluorescence, osmotic adjustment, lipid peroxidation and loss of selective permeability of membranes). The two species tested in this study have revealed great potential for removing toxic heavy metals and filtering polluted wastewater. Our results show the performance of *S. alba* seedlings for rhizofiltration applications with a preferential accumulation of heavy metals in senescent older leaves. In addition, *S. alba* seedlings showed a recovery period with the emergence of new leaves during the experiment, allowing better physiological functioning. The results make it possible to continue this study on a larger operational scale in situ by associating the two species and applying the treatment over a longer period.

**Author Contributions:** Conceptualization, M.B. and Z.B.; Methodology, experimental design, data collection, laboratory analyses: M.B., Z.B., M.S.L., M.A. and D.P.K.; Statistical analyses: M.B.; writing—original draft preparation, M.B., Z.B. and M.S.L.; writing—review and editing, M.B., Z.B., M.S.L., M.A. and D.P.K. All authors have read and agreed to the published version of the manuscript.

**Funding:** This research was funded by the University of Carthage, Ministry of Higher Education and Scientific Research, Tunisia, the National Institute of Research in Rural Engineering, Water and Forests (INRGREF), Tunisia, and the NSERC Discovery Grant to DPK to cover publication costs.

**Data Availability Statement:** Not applicable.

**Acknowledgments:** We are thankful to William F.J. Parsons for English editing. We would also like to extend our thanks to the technicians of the forest ecology laboratory of INGRREF for their valuable technical support to this project.

**Conflicts of Interest:** The authors declare no conflict of interest.

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
