# Peer review of "Potential Use of Two Forest Species (Salix alba and Casuarina glauca) in the Rhizofiltration of Heavy-Metal-Contaminated Industrial Wastewater"

_forests, doi:10.3390/f14030654_

Round 1

Reviewer 1 Report

I have carefully reviewed this manuscript. I have many comments/remarks on this manuscript. In my opinion, manuscript can be accepted after major revision.

Line 3: “indus-trial” - please write in one line

Line 33-34: I recommend modifying the sentence or deleting it.

Line 41-42: It has been widely reported that IWW often contains high concentrations of heavy metals [3]. I suggest adding some data on the content of heavy metals in certain types of industrial waters (from electroplating, tannery, etc.), while also pointing out which metals are generally present in industrial wastewater.

Line 56-59: I suggest to mention that in general, conventional industrial water treatment methods do not show high efficiency when heavy metals are in lower concentrations (e.g. < 100 mg/L). Furthermore bioremediation methods have been shown to be effective when heavy metals are found in lower concentrations. There are many studies in the literature that refer to these issues.

Line 89-90: removal efficiencies for what concentration of metal in the environment were recorded?

Line 122: Preliminary tests are done in order to establish the concentration limits in which rhizofiltration can be applied. But from the data presented in the paper it appears that the experiments started considering 0, 25, 50, 75, 100% IWW, and the results showed that the concentration that exceeds 25%, IWW is lethal to the seedlings. In the manuscript you present data for 0%IWW (which is the control) and for 25%IWW. No results are presented at intermediate concentrations. So, I suggest rewriting the sentence.

Line 321. Try to fit the 6 figures forming figure 3 on one page. The same observation for figure 4.

Line 390. Rebuild table 2 so that it fits on the portrait page and not on the landscape page.

Avoid empty pages inside the manuscript

Line 406. Consider “Morphological changes and seedling growth” as 4.1. subtitle.

Line 433: Consider “Water status, chlorophyll concentration and chlorophyll fluorescence” as 4.2. subtitle. Consider this observation and for lines 459 and 475.

Line 437: Beyer et al.

Line 510: Beyer et al.

The references in text should be cited using the abbreviation “et al.” and not “and al.”. Please correct in the whole manuscript.

I suggest to be consulted the instructions for authors regarding manuscript writing. There are parts that do not comply with the instructions specified by the journal. 

Reviewer 2 Report

In this study, two plants, Salix alba and Casuarina glauca, were used for the remediation of industrial wastewater, and good remediation results were achieved. The results showed that Salix alba has a strong ability to remove heavy metal ions and can be used as an effective remediation agent for toxic metal-polluted industrial wastewater. The measurement indexes are relatively abundant and persuasive. It is recommended to be accepted after minor revision, and there are some specific issues below, which are suggested to be revised.

1. the abstract section lacks specific experimental data to support the conclusions of the article, and it is suggested to be added.

2. In the preface section, no specific scientific question is seen and a detailed elaboration is suggested.

3. line 277, the results section cannot have a discussion section and it is suggested to move to the discussion section later.

4. the discussion section lacks the experimental results of this study, and it is suggested to add some data from the experiments and then compare and discuss them.

5. The conclusion section also does not see the specific experimental data, and it is suggested to make appropriate additions.

Round 2

Reviewer 1 Report

After reading the manuscript, I still have same observations:

Line 128-130: I recommend not using the term “Preliminary tests” to avoid confusion about experiments.

My recommendation is:

“Five different concentrations of IWW (0, 25, 50, 75, 128 100% IWW) were used to evaluate the ecotoxicity of industrial wastewater and its effect on the ecophysiology Casuarina glauca and Salix alba, but the concentration higher than 25% IWW was found to be lethal to seedlings.”

Tabeles 2 and 3 I recommend that the values in the tables have the same number of decimal places. For example the content of Na in C is 0.470 but in S is 0.44. Maintain uniformity in data presentation.
